

# The spin Drude weight of the XXZ chain and generalized hydrodynamics

Andrew Urichuk[1,2], Yahya Öz[1], Andreas Klümper[1] and Jesko Sirker[2*]

**1** Fakultät für Mathematik und Naturwissenschaften, Bergische Universität Wuppertal, 42097 Wuppertal, Germany
**2** Department of Physics and Astronomy, University of Manitoba, Winnipeg R3T 2N2, Canada

* sirker@physics.umanitoba.ca

## Abstract

Based on a generalized free energy we derive exact thermodynamic Bethe ansatz formulas for the expectation value of the spin current, the spin current-charge, charge-charge correlators, and consequently the Drude weight. These formulas agree with recent conjectures within the generalized hydrodynamics formalism. They follow, however, directly from a proper treatment of the operator expression of the spin current. The result for the Drude weight is identical to the one obtained 20 years ago based on the Kohn formula and TBA. We numerically evaluate the Drude weight for anisotropies $\Delta = \cos(\gamma)$ with $\gamma = \pi n/m$, $n \leq m$ integer and coprime. We prove, furthermore, that the high-temperature asymptotics for general $\gamma = \pi n/m$—obtained by analysis of the quantum transfer matrix eigenvalues—agrees with the bound which has been obtained by the construction of quasi-local charges.

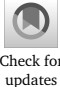

## 1  Introduction

The Hamiltonian of the XXZ chain is given by

$$H = J \sum_{l=1}^{N} \left( \sigma_l^x \sigma_{l+1}^x + \sigma_l^y \sigma_{l+1}^y + \Delta \sigma_l^z \sigma_{l+1}^z \right) - 2h \sum_{l=1}^{N} \sigma_l^z, \tag{1}$$

where $\sigma^{x,y,z}$ are Pauli matrices, $\Delta = \cos(\gamma)$ is the anisotropy, $h$ the applied magnetic field, and we use periodic boundary conditions. The XXZ chain is a Bethe ansatz (BA) integrable model and a family of commuting transfer matrices, $[T(\theta), T(\theta')] = 0$, exists with $\theta$ being the spectral parameter. The logarithm of the transfer matrix is the generating function for an infinite set of conserved charges

$$Q_n = \frac{d^n}{d\theta^n} \ln T(\theta) \bigg|_{\theta=0}. \tag{2}$$

In particular, $Q_1 \propto H$ and $Q_2 \propto J_E$ for zero magnetic field, where $J_E$ is the energy current operator. Based on the infinite number of conservation laws, one might expect that the XXZ chain shows purely ballistic transport. This is indeed the case for thermal transport because the energy current $J_E$ is itself a conserved charge, i.e. $[J_E, H] = 0$. Based on a generalized Gibbs ensemble (GGE), which includes the higher conserved charges, the temperature dependence of the thermal conductivity can thus be calculated straightforwardly [1,2].

The spin current operator $J_0$, on the other hand, is not conserved. Whether the spin Drude weight is finite at finite temperatures and, if so, how to calculate it analytically has been the subject of a number of studies in the last 20 years [3–15]. Based on a field theoretical treatment, a coexistence of a Drude weight with a diffusive part was predicted for small finite temperatures [7]. In frequency space, this corresponds to a Drude peak which sits on top of a narrow Lorentzian. Further evidence for this picture was recently obtained in a generalized hydrodynamics equation where a diffusive term was considered as next leading correction [16]. In this paper we will not study the diffusive part of the current and will instead be exclusively concerned with the calculation of the ballistic part, i.e. the finite-temperature Drude weight.

The spin current density is defined by the discrete continuity equation

$$\partial_t \sigma_l^z = -\mathrm{i}[\sigma_l^z, H] = -(j_l - j_{l-1}), \tag{3}$$

from which one obtains

$$j_l = 4iJ \left( \sigma_l^+ \sigma_{l+1}^- - \sigma_l^- \sigma_{l+1}^+ \right), \tag{4}$$

with $\sigma^{\pm} = (\sigma^x \pm i\sigma^y)/2$. The total spin current operator is given by $J_0 = \sum_l j_l$. The spin Drude weight $D(\beta)$ at inverse temperature $\beta = 1/T$ (we set $k_B = 1$) can then be defined in the following two equivalent ways. On the one hand, one can consider the Kubo formula for the spin conductivity as a function of frequency $\omega$

$$\sigma(\omega) = \frac{\mathrm{i}}{\omega} \left[ \frac{\langle H_{\mathrm{kin}} \rangle}{N} + \langle J_0 ; J_0 \rangle_{\mathrm{ret}}(\omega) \right], \tag{5}$$

where $H_{\text{kin}}$ is the kinetic energy operator, i.e. the transversal exchange terms of $H$, and $\langle \ ; \ \rangle_{\text{ret}}$ is the retarded correlation function. The real part of the conductivity is given by

$$\sigma'(\omega) = 2\pi D(\beta)\delta(\omega) + \sigma_{\text{reg}}(\beta, \omega). \tag{6}$$

A finite Drude weight, $D(\beta) > 0$, thus implies an infinite dc conductivity. Another way to define the Drude weight is to consider the current-current correlator directly in time $t$,

$$D(\beta) = \lim_{t\to\infty} \lim_{N\to\infty} \frac{\beta\langle J_0(0)J_0(t)\rangle}{2N} = \lim_{N\to\infty} \frac{\beta}{2N} \sum_k \frac{|\langle J_0 Q_k\rangle|^2}{\langle Q_k^\dagger Q_k\rangle}. \tag{7}$$

In the last step, we have projected onto a *complete* set of commuting conserved charges which are orthogonal, $\langle Q_k^\dagger Q_l\rangle = \langle Q_k^\dagger Q_k\rangle\delta_{kl}$ where $\langle\ldots\rangle$ denotes the thermal average at inverse temperature $\beta$. If the set of charges is not complete then the r.h.s. provides a lower bound for $D(\beta)$, the so-called Mazur bound [3,17,18]. According to this equivalent definition, the Drude weight is the part of the current which does not decay in time because it is protected by a finite overlap with some of the conserved charges. The question of whether or not the XXZ chain always has a finite Drude weight at finite temperatures is an intriguing one because $\langle J_0 Q_k\rangle = 0$ for all the charges defined in Eq. (2) if the magnetic field vanishes. This follows from simple symmetry considerations: While the spin current (4) is odd under the spin-flip symmetry $\sigma^z \to -\sigma^z$, all the charges in Eq. (2) are even in this case. This puzzle was solved by realizing that for the open XXZ chain, additional operators exist which are conserved up to boundary terms and are odd under the spin-flip operation [8,9]. Later it was shown that fully conserved odd charges with finite overlap with the current operator can be constructed for periodic boundary conditions [10,11]. In the following, we concentrate exclusively on the Drude weight for vanishing magnetic field, h=0.

Using the Bethe ansatz there are three different approaches which have been used so far to compute the Drude weight: (1) Starting from the spectral representation of the Kubo formula (5) and comparing this with the change of the eigenenergies $\varepsilon_n$ of the Hamiltonian (1) when threading a static magnetic flux $\Phi$ through an XXZ ring one finds

$$D = \frac{1}{2NZ} \sum_n e^{-\beta\varepsilon_n} \left.\frac{\partial^2 \varepsilon_n(\Phi)}{\partial\Phi^2}\right|_{\Phi=0}, \tag{8}$$

with $Z$ the partition function. This is a generalization of the Kohn formula [19] to finite temperatures [20]. For zero temperature, in particular, the Drude weight can be obtained simply from the ground state energy of the system with an added flux [21] leading to

$$\lim_{\beta\to\infty} D(\beta) = D_{\beta\to\infty} = J\frac{\pi\sin\gamma}{2\gamma(\pi-\gamma)}. \tag{9}$$

For finite inverse temperatures, the formula (8) has been used in Ref. [4] to calculate $D(\beta)$ for anisotropies $\gamma = \pi/m$ on the basis of the thermodynamic Bethe ansatz (TBA). The high- and low-temperature limits have then been analyzed in Ref. [5]. (2) A completely different approach is based on constructing a set of charges that have finite overlap with the current operator and to evaluate the r.h.s. of Eq. (7), see Refs. [8–11]. A major difficulty in this approach is the evaluation of the correlators at finite temperatures. So far, only the high-temperature limit has been analyzed analytically [9] resulting in

$$\lim_{\beta\to 0} \frac{4}{\beta} D = J^2 \frac{\sin^2(\pi n/m)}{\sin^2(\pi/m)} \left(1 - \frac{m}{2\pi}\sin(2\pi/m)\right). \tag{10}$$

Here the equal sign is only correct if the set of conserved charges used is complete which is a point which is difficult to prove. For anisotropies $\gamma = \pi/m$ it has been shown that the above

result agrees with the high-temperature limit of the TBA result obtained using the Kohn formula. In the following we will prove that this is also true for general anisotropies $\gamma = \pi n/m$. Note that the Drude weight in the high-temperature limit has a fractal character according to Eq. (10) while $D_{\beta \to \infty}$ depends smoothly on anisotropy, see Eq. (9). Finally (3), a third approach has recently been proposed based on a generalized hydrodynamics (GHD) formulation where it is conjectured that the continuity equation

$$\partial_t \langle q_\ell \rangle + \partial_x \langle j_\ell \rangle = 0 \tag{11}$$

takes the form of an Euler equation[1] for the $\ell$-th quasi-particle density $\rho_\ell$ [12–15]

$$\partial_t \rho_\ell(\theta) + \partial_x (v_\ell(\theta)\rho_\ell(\theta)) = 0, \tag{12}$$

with effective velocity $v_\ell(\theta)$, where we have suppressed the time and space dependence. It should also be noted that both the particle density and effective velocity depend on position and time. The expectation value of an extensive charge $\langle Q_m \rangle$ in a local stationary state described by the distribution $\rho_\ell$ is given by

$$\langle \rho | Q_m | \rho \rangle / N = \sum_\ell \int d\theta \, q_\ell^m(\theta) \rho_\ell(\theta), \tag{13}$$

with the subscript referring to an $\ell$-string in the BA solution and with the superscript denoting the $m$–th bare charge eigenvalue. If one assumes that a system which is not in equilibrium is composed of cells which are locally described by the distribution $\rho_\ell(\theta)$ then Eq. (12) allows to compute the time evolution of the system along every ray $\xi = x/t$. For the Lieb-Liniger model in the linear response regime, in particular, this formalism has been used to obtain formulas for the expectation values of $\langle J_n \rangle$, $\langle J_n Q_m \rangle$, and $\langle Q_n Q_m \rangle$ [15]. Formally, these results can be straightforwardly generalized to the XXZ chain by summing over all possible string types. An obvious question then is if the TBA formulas for the current and current-charge expectation values obtained in this way are exact.

To answer this question we will present in this paper a fourth approach where we derive current and current-charge correlators exactly starting from a generalized free energy and the operator expression for the spin current, without using the GHD conjecture. Based on Eq. (7) we will then use these correlators to derive a formula for the Drude weight and show that it is identical to the GHD result and to the TBA result obtained from the Kohn formula. Our paper is organized as follows: In Sec. 2 we derive exact results for $\langle J_0 \rangle$ and $\langle J_0 Q_n \rangle$. In Sec. 3 we obtain the Drude weight and analyze analytically the high- and low-temperature limits for anisotropies $\gamma = \pi n/m$. A numerical evaluation of the Drude weight for these anisotropies and arbitrary temperatures is presented in Sec. 4. A brief summary and conclusions are given in Sec. 5.

## 2 The spin current, current-charge and charge-charge correlators

The basic object we want to consider is the reduced $n$-site density matrix $D(n)$ obtained from the full thermal density matrix $\rho = \exp(-\beta H)/Z$ by taking a partial trace over the other $N - n$ sites, $D(n) = \text{tr}_{1,\dots,N-n} \rho$. Note that the Hamiltonian (1) is translationally invariant. The reduced density matrix is thus only a function of the length of the segment. The elements of the $n$-site reduced density matrix can always be expressed through a combination of $n$-site spin correlators. For the 1-site reduced DM, for example, we have $1 = D_1^1 + D_2^2$,

---

[1]This equation is often referred to as the Bethe-Boltzmann equation.

$\langle \sigma^z \rangle = \text{tr}(D(1)\sigma^z) = D_1^1 - D_2^2$, $\langle \sigma^+ \rangle = D_1^2$, and $\langle \sigma^- \rangle = D_2^1$ which allows to rewrite the matrix elements $D_\alpha^\beta$ in terms of the expectation values of $\sigma^{z,+,-}$. Similarly, for the 2-site reduced DM we find $D_{12}^{21} = \langle \sigma_l^+ \sigma_{l+1}^- \rangle$. Knowing the elements of the 2-site density matrix thus allows to determine the expectation value of the spin current operator defined in Eq. (4). Using the Yang-Baxter algebra, the following relation for an inhomogeneous generalization of the reduced density matrix has already been obtained previously [22]

$$D_{12}^{21}(2;\xi_1,\xi_2) - D_{21}^{12}(2;\xi_2,\xi_1) = \frac{D_1^1(1;\xi_1) - D_1^1(1;\xi_2)}{i(\xi_1 - \xi_2)}. \tag{14}$$

Here $\xi_i$ are spectral parameters which are put on the vertical lines of the corresponding vertex model. Identifying the matrix elements by the spin correlators as above we find from (14) the relation

$$\langle j_l \rangle = 2\partial_\xi \langle \sigma_l^z \rangle \big|_{\xi=0} \quad ; \quad \langle J_0 \rangle / N = \langle j_l \rangle = -\partial_\xi \partial_{\beta h} f_\xi(\{\beta\}) \big|_{\xi,h=0}. \tag{15}$$

Here $f_\xi(\{\beta\})$ is the generalized free energy density with spectral parameter $\xi$ and generalized inverse temperatures $\{\beta\} = \{\beta_0, \beta_1, \dots\}$. It is related to the leading eigenvalue $\Lambda(\xi)$ of the quantum transfer matrix by

$$f_\xi(\{\beta\}) = -\ln \Lambda(\xi). \tag{16}$$

We want to stress once more that we discuss here only the transport properties of the XXZ chain at zero magnetic field. In TBA we can write this free energy density as

$$f_\xi(\{\beta\}) = -\frac{1}{2\pi} \sum_\ell \int d\theta\, p_\ell'(\xi - \theta)\sigma_\ell \ln[1 + \eta_\ell^{-1}(\theta)]. \tag{17}$$

Here $p_\ell(\theta)$ is the momentum distribution and the variables $\sigma_\ell = \text{sign}(g_\ell)$ are the signs of auxiliary rational numbers associated to string solutions as defined in [23]. For the simplest case of anisotropy $\gamma = \pi/m$ the $g_\ell$ have a particularly simple relation to string length $n_\ell$

$$g_\ell = m - n_\ell, \; n_\ell = \ell \; \text{ for } \ell = 1, \dots, m-1 \text{ and } g_m = -1, n_m = 1. \tag{18}$$

The function $\eta_\ell = \rho_\ell^h / \rho_\ell$ is defined by the ratio of hole density $\rho_\ell^h$ and particle density $\rho_\ell$ of the $\ell$-th particle (string). It fulfills the TBA equations

$$\ln \eta_\ell(\theta) = \sum_n \beta_n q_\ell^n(\theta) + \sum_\kappa \int d\mu K_{\ell\kappa}(\theta - \mu)\sigma_\kappa \ln(1 + \eta_\kappa^{-1}(\mu))$$
$$\equiv \sum_n \beta_n q_\ell^n + \left[ K * \sigma \ln(1 + \eta^{-1}) \right]_\ell, \tag{19}$$

with charges $q_\ell^n$, Lagrange multipliers (generalized temperatures) $\beta_n$, an integration kernel $K$, and '$*$' denoting a convolution and sum over Bethe strings. For the first few charges we have, in particular, $\beta_0 = \beta h$, $q_\ell^0 = n_\ell$, and $\beta_1 = \beta$, $\gamma(4J\sin\gamma)^{-1}\varepsilon_\ell = \partial_\theta p_\ell = p_\ell'$. In the following, we rescale the energy $\gamma(4J\sin\gamma)^{-1}\varepsilon_\ell \to \varepsilon_\ell = q_\ell^1$ to absorb the scaling factor. Furthermore, we use the shorthand notation $\partial_n \equiv \partial_{\beta_n}$. Dressed charges $\widetilde{q}_\ell^n$ are defined by the integral relation

$$\widetilde{q}_\ell^n = q_\ell^n - \left[ K * \sigma \vartheta \widetilde{q}^n \right]_\ell, \tag{20}$$

where we have defined the Fermi factor $\vartheta_\ell = 1/(1 + \eta_\ell) = \rho_\ell/(\rho_\ell + \rho_\ell^h)$. It is also very useful to realize the following simple relation of the dressed charges to logarithmic derivatives of the $\eta$-functions

$$\partial_n \log \eta_\ell(\theta) = \widetilde{q}_\ell^n(\theta). \tag{21}$$

In order to calculate the expectation value of the current $\langle J_0 \rangle$ we note that

$$\partial_0 \ln(1 + \eta_\ell^{-1}) = -\frac{\partial_0 \ln \eta_\ell}{1 + \eta_\ell} = -\vartheta_\ell \tilde{q}_\ell^0, \tag{22}$$

leading to

$$\langle J_0 \rangle / N = -\partial_\xi \partial_0 f_\xi(\{\beta\})\big|_{\xi=0} = \frac{1}{2\pi} \sum_\ell \int d\theta \, \sigma_\ell \varepsilon_\ell'(\theta) \vartheta_\ell(\theta) \tilde{q}_\ell^0(\theta), \tag{23}$$

where we have used that $p''(\theta) = \varepsilon'(\theta)$. There are various ways to rewrite this equation. Here we want to bring it into a form similar to the one conjectured within the GHD approach. The basic identity we want to make use of is

$$\sum_\ell \int d\theta \left[ K * \sigma \partial_n \ln(1 + \eta^{-1}) \right]_\ell \sigma_\ell \partial_m \ln(1 + \eta_\ell^{-1})$$
$$= \sum_\ell \int d\theta \left[ K * \sigma \partial_m \ln(1 + \eta^{-1}) \right]_\ell \sigma_\ell \partial_n \ln(1 + \eta_\ell^{-1}). \tag{24}$$

Using Eq. (19) we can express $K * \sigma_\ell \partial_0 \ln(1 + \eta_\ell^{-1}) = \partial_0 \ln \eta_\ell - q_\ell^0$ and $K * \sigma_\ell \partial_2 \ln(1 + \eta_\ell^{-1}) = \partial_2 \ln \eta_\ell - q_\ell^2$. In this case the identity (24) yields

$$\sum_\ell \int d\theta \, q_\ell^0 \sigma_\ell \underbrace{\partial_2 \ln(1 + \eta_\ell^{-1})}_{-\vartheta_\ell \tilde{q}_\ell^2} = \sum_\ell \int d\theta \, q_l^2 \sigma_\ell \underbrace{\partial_0 \ln(1 + \eta_\ell^{-1})}_{-\vartheta_\ell \tilde{q}_\ell^0}. \tag{25}$$

The expectation value of the current operator (23) can thus also be written as

$$\langle J_0 \rangle / N = \frac{1}{2\pi} \sum_\ell \int d\theta \, \sigma_\ell \vartheta_\ell \underbrace{\tilde{q}_\ell^2}_{=\widetilde{\partial_\theta \varepsilon_\ell}} \underbrace{q_\ell^0}_{=n_\ell} = \sum_\ell \int d\theta \, v_\ell(\theta) \rho_\ell(\theta) q_\ell^0(\theta), \tag{26}$$

where the rapidity density $\widetilde{\partial_\theta p_\ell} = 2\pi \sigma_\ell(\rho_\ell + \rho_\ell^h)$ and effective velocity $v_\ell \equiv \widetilde{\partial_\theta \varepsilon_\ell} / \widetilde{\partial_\theta p_\ell}$ are defined by the dressed derivatives with respect to the spectral parameter of the energy and the momentum. This formula agrees with the conjectured general current formula used in GHD and appearing in Ref. [12, 13].

The correlator $\langle J_0 Q_n \rangle / N$ can also be computed from the free energy $f_\xi(\{\beta\})$ defined in Eq. (17) via derivatives with respect to the appropriate Lagrange multiplier $\beta_j$, see Eq. (21). We find in particular,

$$\langle J_0 Q_n \rangle / N = -\partial_\xi \partial_0 \partial_n f_\xi(\{\beta\}) = -\frac{1}{2\pi} \sum_\ell \int d\theta \, (\partial_\theta \varepsilon_\ell) \sigma_\ell \partial_0 \partial_n \ln(1 + \eta_\ell^{-1}). \tag{27}$$

In order to simplify this result we use the following relation

$$\sum_\ell \int d\theta \, q_\ell^k \sigma_\ell \partial_m \partial_n \ln(1 + \eta_\ell^{-1}) = \sum_\ell \int d\theta \, \sigma_\ell \vartheta_\ell(1 - \vartheta_\ell) \tilde{q}_\ell^k \tilde{q}_\ell^m \tilde{q}_\ell^n, \tag{28}$$

which is proven in Appendix A. Using this relation for the charge-current correlator (27) leads to our final result

$$\langle J_0 Q_n \rangle / N = -\sum_\ell \int \frac{d\theta}{2\pi} \widetilde{\partial_\theta \varepsilon_\ell} \sigma_\ell \vartheta_\ell(1 - \vartheta_\ell) \tilde{q}_l^0 \tilde{q}_\ell^n = -\sum_\ell \int d\theta \, v_\ell \rho_\ell(1 - \vartheta_\ell) \tilde{q}_\ell^0 \tilde{q}_\ell^n, \tag{29}$$

where we have once more made use of the rapidity density and effective velocity relations. As above, this result is consistent with a generalization of the formula in Ref. [15] from the Lieb-Liniger model to the case of multiple particle species. Analogously, the charge-charge correlator is given by

$$
\begin{aligned}
\langle Q_n Q_m \rangle / N &= -\partial_n \partial_m f_{\xi=0}(\{\beta_n\}) = \frac{1}{2\pi} \sum_\ell \int d\theta \, (\partial_\theta p_\ell) \sigma_\ell \partial_n \partial_m \ln(1 + \eta_\ell^{-1}), \\
&= \frac{1}{2\pi} \sum_\ell \int d\theta \, \sigma_\ell \vartheta_\ell (1 - \vartheta_\ell) \widetilde{\partial_\theta p_\ell} \widetilde{q}_\ell^{\,n} \widetilde{q}_\ell^{\,m}, \\
&= \sum_\ell \int d\theta \, \rho_\ell (1 - \vartheta_\ell) \widetilde{q}_\ell^{\,n} \widetilde{q}_\ell^{\,m},
\end{aligned}
\tag{30}
$$

where we have used again the relation (28) in the second step. For the special case $Q_n = Q_m = Q_2 = J_E$ this reproduces the formula needed to calculate the thermal Drude weight by TBA first derived in [2]. If we only take a single derivative, then we obtain a TBA formula for the energy current

$$
\langle J_E \rangle = -\partial_2 f_{\xi=0}(\{\beta\}) = \frac{1}{2\pi} \sum_\ell \int d\theta \, p'_\ell \sigma_\ell \vartheta_\ell \tilde{q}_\ell^{\,2} = \sum_\ell \int d\theta \, \rho_\ell q_\ell^2 = \sum_\ell \int d\theta \, v_\ell \rho_\ell q_\ell^1.
\tag{31}
$$

This result agrees with Eq. (13) and also with Eq. (26) provided we replace $q_\ell^0$ with $q_\ell^1$.

## 3 The Drude weight

Using the expressions for the spin-charge and charge-charge correlators in Eq. (29) and Eq. (30) of the previous section, the Drude weight (7) can be determined from

$$
D = \lim_{N \to \infty} \frac{\beta}{2N} \sum_n \frac{\langle J_0 Q_n \rangle^2}{\langle Q_n^2 \rangle} = \frac{\beta}{2} \sum_n \frac{\left( \sum_\ell \int d\theta \, v_\ell \rho_\ell (1 - \vartheta_\ell) \widetilde{q}_\ell^{\,0} \widetilde{q}_\ell^{\,n} \right)^2}{\sum_\ell \int d\theta \, \rho_\ell (1 - \vartheta_\ell) \widetilde{q}_l^{\,n} \widetilde{q}_\ell^{\,n}}.
\tag{32}
$$

In order to simplify Eq. (32) we follow the original argument by Mazur [17] and define a quantity $Z = J_0 - \sum_n c_n Q_n$ with $\langle Z^2 \rangle \geq 0$, and the set $Q_n$ being the complete set of conserved charges. This leads to the relation[2]

$$
\langle J_0 J_0 \rangle \geq 2 \sum_n c_n \langle J_0 Q_n \rangle - \sum_{n,m} c_n c_m \langle Q_n Q_m \rangle.
\tag{33}
$$

Maximizing the right hand side with respect to the vector $\vec{c}$ leads to the condition

$$
\sum_n c_n \langle Q_n Q_m \rangle = \langle J_0 Q_m \rangle.
\tag{34}
$$

Using the expressions (29) and (30) after bringing the overall constants to the LHS

$$
\begin{aligned}
&\sum_{n,\ell} c_n \int d\theta \, \rho_\ell(\theta) (1 - \vartheta_\ell(\theta)) \widetilde{q}_\ell^{\,n}(\theta) \widetilde{q}_\ell^{\,m}(\theta) = -\sum_\ell \int d\theta \, v_\ell(\theta) \rho_\ell(\theta) (1 - \vartheta_\ell(\theta)) \widetilde{q}_\ell^{\,0} \widetilde{q}_\ell^{\,m}(\theta) \\
\Leftrightarrow \quad &\sum_\ell \int d\theta \, \rho_\ell(\theta) (1 - \vartheta_\ell(\theta)) \left( \sum_n c_n \widetilde{q}_\ell^{\,n}(\theta) + v_\ell(\theta) \widetilde{q}_\ell^{\,0} \right) \widetilde{q}_\ell^{\,m}(\theta) = 0.
\end{aligned}
\tag{35}
$$

---

[2]Taking $\langle J_0 J_0 \rangle$ as shorthand for $\lim_{t \to \infty} \langle J_0(0) J_0(t) \rangle$.

Next, we use that $\{q^n\}$ is a complete set of conserved charges with non-vanishing overlap with the spin current. This set comprises of the quasi-local charges [9] and additional charges $Q_1, Q_2, \cdots$. The exact form of these additional charges does not matter as long as they make the set complete. If this is the case, then the Mazur argument can be applied. We will see in the following that the additional charges drop out in the final result. We further also assume completeness in the sense that the vanishing of the sum-integral of Eq. (35) for any charge automatically implies the vanishing of the integrand,

$$\sum_n c_n \widetilde{q}_\ell^n(\theta) = -\nu_\ell(\theta)\widetilde{q}_\ell^0. \tag{36}$$

Under these assumptions, the bound obtained should be exhaustive and we find the following expression for the conserved part of the spin current

$$
\begin{aligned}
2D\beta^{-1} = \langle J_0 J_0\rangle/N &= \sum_n c_n \langle J_0 Q_n\rangle/N - \sum_{n,m} c_n c_m \langle Q_n Q_m\rangle/N = \sum_n c_n \langle J_0 Q_n\rangle/N \\
&= -\sum_\ell \int d\theta \sum_n c_n \rho_\ell(\theta)(1-\vartheta_\ell(\theta))\widetilde{q}_\ell^n(\theta)\,\nu_\ell(\theta)\widetilde{q}_\ell^0(\theta) \\
&= \sum_\ell \int d\theta\, \rho_\ell(\theta)(1-\vartheta_\ell(\theta))\big[\nu_\ell(\theta)\widetilde{q}_\ell^0(\theta)\big]^2.
\end{aligned}
\tag{37}
$$

The last part of our derivation is based on the same assumptions used in [15]. Importantly however, the expressions for current and current-charge expectation values are derived from first principles.

## 3.1   Equivalence with the Drude weight formula by Zotos

Starting from (37) it is now straightforward to show that our result is identical to the one obtained 20 years ago based on the Kohn formula and using the TBA to calculate the curvature of energy levels [4,5]. Rewriting the particle density and filling fraction in terms of $\eta$-functions we obtain

$$
\begin{aligned}
D &= \frac{\beta}{2}\sum_\ell \int d\theta\, \frac{\rho_\ell + \rho_\ell^h}{(1+\eta_\ell)(1+\eta_\ell^{-1})}\left(\widetilde{\partial_\theta \varepsilon}/\widetilde{\partial_\theta p}\right)^2 (\widetilde{q}_\ell^0)^2 \\
&= \frac{\beta}{4\pi}\sum_\ell \int d\theta\, \sigma_\ell \frac{(\widetilde{q}_\ell^2)^2 (\widetilde{q}_\ell^0)^2}{\widetilde{q}_\ell^1 (1+\eta_\ell)(1+\eta_\ell^{-1})}.
\end{aligned}
\tag{38}
$$

Now we can use the relation $\partial_\theta \ln \eta_\ell = \beta\widetilde{\partial_\theta \varepsilon}_\ell = \beta\widetilde{q}_\ell^2$ to obtain—up to a normalization factor—the well-known result [5]

$$D = \frac{1}{4\pi\beta}\sum_\ell \int d\theta\, \sigma_\ell \frac{(\partial_\theta \ln \eta_\ell)^2 (\partial_0 \ln \eta_\ell)^2}{(\partial_1 \ln \eta_\ell)(1+\eta_\ell)(1+\eta_\ell^{-1})}. \tag{39}$$

Note that by restoring the scaling factor $4J\sin(\gamma)/\gamma$ the result in (39) agrees with [5].

Here we have thus provided an alternative derivation of the Drude formula which makes use of an (exhaustive) Mazur bound and first principles derivations of current correlators instead of the Kohn formula. Note, however, that both approaches use the TBA formalism so the rederivation presented here should not be understood as being completely independent.

### 3.2 Low-temperature limit

The low-temperature asymptotics of Eq. (39) have already been determined in [21, 24, 25] with

$$D_{\beta \to \infty} = J \frac{\pi \sin \gamma}{2\gamma(\pi - \gamma)}, \tag{40}$$

consistent with the known zero temperature result (9). Our numerical data discussed in more detail in Sec. 4 also agree with this low-temperature formula, up to the point where the numerics breaks down, see Fig. 3.

We note, furthermore, that this formula also follows directly from the alternative expression (37) by observing that the particle/ hole densities vanish around the origin. So only regions with constant effective velocity $v_{\pm} = \pm 2J\pi \sin(\gamma)/\gamma$ have non-zero particle/ hole density. Then taking into account that (29) reduces to $(v_{\pm})^2$ multiplied by $1/2$ times the zero field susceptibility, $\chi_0 = \frac{1}{4J\pi(\pi-\gamma)} \frac{\gamma}{\sin \gamma}$, Eq. (40) follows provided that one also reintroduces the rescaling factor $4J \sin(\gamma)/\gamma$.

### 3.3 High-temperature limit

The high temperature asymptotics of the Drude weight (10) has been obtained by constructing families of quasi-local charges [9]. Numerics based on the GHD approach agree with this bound [28] and analytical GHD calculations for certain density and current profiles reproduce it [26]. A proof for (10) directly from the quantum transfer matrix approach is known only for $\gamma = \frac{\pi}{m}$ [5]. We generalize this transfer matrix result to anisotropies $\gamma = \frac{n\pi}{m}$ making use of the Y-system decomposition in [27], and the usual unscaled temperatures appearing therein. A rational $\pi/\gamma$ can be written as a continued fraction of length $\alpha$ determined by integers $v_j$. There are $L = \sum_{j=1}^{\alpha} v_j$ functional equations for $\eta_\ell$ terms. Importantly the final two 'boundary' $\eta$ are given by

$$\eta_{L-1}(x) = e^{\beta h m/2} K(x), \qquad \eta_L(x) = e^{\beta h m/2} \frac{1}{K(x)}. \tag{41}$$

These boundary $\eta$ are the only terms with magnetization appearing in odd powers, meaning that

$$\partial_{\beta h} \eta_{L-1}\big|_{h=0} = \partial_{\beta h} \eta_L\big|_{h=0} = \frac{m}{2}, \quad \text{and} \quad \partial_{\beta h} \eta_j\big|_{h=0} = 0, \quad \text{for} \quad 1 \le j \le L-2. \tag{42}$$

Thus only the final two $\eta$ terms contribute to the Drude weight at vanishing field. We denote the boundary string pair as a particle ($\eta_{L-1}$)/ hole ($\eta_L$) pair with string lengths $\mu := n_{L-1}$ and $\bar\mu := n_L$ respectively. From Eq. (41), this pair is determined by $K(x)$, with these $K(x)$ expressible in terms of transfer matrices $T_{r-1}(x)$,

$$K(x) = \frac{T_{\mu-1}(x + ip_0 w_0)}{T_{\bar\mu-1}(x + i(m + p_0 w_0))}, \tag{43}$$

with $w_0 p_0 = (-1)^{\alpha+1} p_L + p_0 - 2\bar\mu$, where $\alpha$ is the length of our continued fraction, $p_0 = \pi/\gamma$, and $p_L = \pi/(\gamma m)$. By abuse of notation we express the eigenvalues of $T_{r-1}$ as

$$Q(x) = \prod_{j=1}^{M} \sinh\left(\frac{\gamma}{2}(x - \omega_j)\right), \tag{44}$$

$$\phi^{\pm}(x) = \left\{ \sinh\left(\frac{\gamma}{2}(x \pm iu)\right) \right\}^{\frac{N}{2}}, \text{ where } u = -\frac{4J\beta \sin(\gamma)}{\gamma N}, \tag{45}$$

$$T_{r-1}(x) = Q(x + ir)Q(x - ir) \sum_{j=1}^{r} \frac{\phi^-[x + i(2j-2-r)]\phi^+[x + i(2j-r)]}{Q[x + i(2j-2-r)]Q[x + i(2j-r)]}. \tag{46}$$

By use of both the functional relation, which is valid for rational values of $\pi/\gamma$,

$$T_{\mu+2\bar{\mu}-1}(x) = T_{\mu-1}(x) + 2T_{\bar{\mu}-1}(x + i(\mu + \bar{\mu})), \tag{47}$$

and periodicity conditions of $Q(x)$ and $\phi(x)$ the sums of Eq. (46) inserted into Eq. (43) simplify to

$$K(x) + 1 = m \left( \sum_{j=1}^{\bar{n}} \frac{\phi^-[x + i(p_0 w_0 + 2j - 2)]\phi^+[x + i(p_0 w_0 + 2j)]}{Q[x + i(p_0 w_0 + 2j - 2)]Q[x + i(p_0 w_0 + 2j)]} \right)^{-1}. \tag{48}$$

In this form the Trotter limit at infinite temperature $\beta \to 0$ can be used to determine the first order temperature effect, by noting that in this limit the Bethe roots can be identified identically with zero. For brevity take $\xi_j(x) = \coth\left(\frac{\gamma x}{2} + i\gamma j\right)$ and the first order in $\beta$ yields (with $x' = x + i w_0 p_0$)

$$\frac{\phi^-[x' + i(2j-2)]\phi^+[x' + i(2j)]}{Q[x' + i(2j-2)]Q[x' + i(2j)]} = \left(1 + iJ \sin\gamma\beta \left(\xi_{j-1}(x') - \xi_j(x')\right)\right). \tag{49}$$

From this expansion it is straightforward to complete the sum in the denominator of Eq. (48). Expanding again in $\beta$ leads to the first order correction

$$K(x) + 1 = \frac{m}{\bar{\mu}} \left( 1 - \frac{iJ \sin\gamma}{\bar{\mu}}\beta \left(\xi_0(x + i w_0 p_0) - \xi_{\bar{\mu}}(x + i w_0 p_0)\right) \right) + O(\beta^2). \tag{50}$$

This result can then be inserted into the Drude weight formula Eq. (39), which reduces to the integral

$$D_{\beta\to 0} = -\frac{i\beta J^2 \sin^2(\gamma)}{8\pi} m \int d\theta \left( \frac{(\partial_\theta \xi_0(2\theta/\gamma + i w_0 p_0) - \partial_\theta \xi_{\bar{\mu}}(2\theta/\gamma + i w_0 p_0))^2}{\xi_0(2\theta/\gamma + i w_0 p_0) - \xi_{\bar{\mu}}(2\theta/\gamma + i w_0 p_0)} \right). \tag{51}$$

This can then be integrated to obtain the leading order corrections of the high temperature result

$$4\beta^{-1} D_{\beta\to 0} = J^2 \frac{\sin^2(\gamma)}{\sin^2(\gamma p_\alpha)} \left( \frac{\gamma p_\alpha m}{\pi} - \frac{m}{2\pi} \sin(2\gamma p_\alpha) \right) + O(\beta^2), \tag{52}$$

where the $O(\beta)$ term is found to vanish. This is exactly the Prosen bound (10) as found via the construction of quasi-local charges in [10, 11] provided $p_\alpha = \pi/(\gamma m)$, which is proven in Appendix B by induction.

## 4 Numerical evaluation of $D(\beta)$ for arbitrary temperatures

In order to obtain the Drude weight, two numerical schemes were used. The first was used as a check and involves the preparation of two spin chains at thermal equilibrium with some small magnetic field difference between the two, which are then joined at the origin. The system is evolved via the Euler relations (12), which permit a linear response calculation of the Drude weight. This first method has been applied to this problem previously in Refs. [14, 28]. The second method involves the explicit evaluation of (37), which can be computed much more quickly and was analytically shown in [15] to be equivalent to the first method.

Both methods involved determining the Fermi-weights $\vartheta_\ell(\theta) = \frac{1}{1+\eta_\ell(\theta)}$ via the Yang-Yang method by obtaining the hole/ particle density ratio $\eta_\ell(\theta)$ via Eq. (19). With an initial guess function $M_\ell(\theta)$ the calculation was carried out by simple half step updates until it reached the desired convergence. Explicitly this was carried out by the following steps

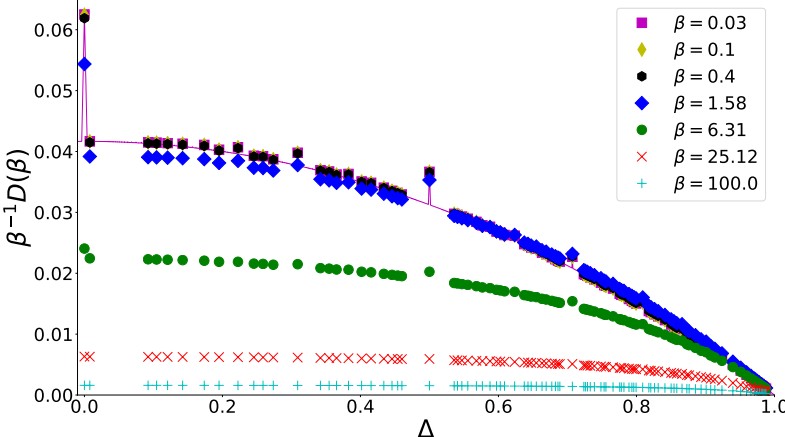

Figure 1: Drude weight coefficient $\beta^{-1}D(\gamma, \beta)$ for various anisotropies $\gamma = \pi n/m$ and temperatures. Note that the high temperature results ($\beta = 0.4, 0.1, 0.03$) are partly on top of each other on this scale and agree with the analytical infinite temperature result (solid line). $D(\beta)$ is a nowhere continuous function except for at $\beta^{-1} = 0$. Note that the change of $\beta^{-1}D$ with decreasing temperature is not uniform: the data for $\beta = 0.4$ and $\beta = 1.58$ show a crossover at $\cos(\gamma) = \Delta \approx 0.59$.

- Use $\eta_\ell^N(\theta)$ to compute the set of $M_\ell^N(\theta) = \log\left(1 + \frac{1}{\eta_\ell^N(\theta)}\right)$.

- Take the Fast Fourier Transform (FFT) of $M_\ell^N(\theta)$.

- Solve the transformed form of Eq. (19) for the dressed energy, $\text{FFT}(\widetilde{\varepsilon}_\ell^{\text{temp}}(\theta))$.

- Invert the FFT and update $\eta_\ell^{N+1} = (\eta_\ell^{\text{temp}} + \eta_\ell^N)/2$.

- Return to the first step with the updated guess $M_\ell^{N+1}$.

Once the functions $\eta_\ell(\theta)$ have converged, the dressed charges can be obtained using the relation (21). The dressed spin is known in the zero field limit to be $\widetilde{q}_\ell^0 = 0$ for $\ell = 1 \ldots L-2$ and $\widetilde{q}_{L-1}^0 = \widetilde{q}_L^0 = m/2$ with $\gamma = \frac{n\pi}{m}$. Note that the $n$ appearing in the anisotropy is not connected to the string length $n_\ell = q_\ell^0$. This provides a first check on the validity of the solution.

A first question we want to address numerically is how the nowhere continuous bound for the Drude weight (10) evolves into the zero temperature Drude weight (9) which is a smooth function of anisotropy. From Fig. 1 it becomes clear that $D(\beta)$ is in fact a fractal for any finite temperature. Once the part of the current which is not protected by conservation laws starts to relax due to finite-temperature Umklapp scattering, the structure of the conserved charges odd under spin-flip symmetry—which strongly depends on the anisotropy $\Delta = \cos(\pi n/m)$—becomes visible in the remaining Drude weight. The $\beta^{-1} = 0$ case is special because Umklapp scattering is an irrelevant operator. There is no mechanism for current relaxation in a completely clean system at zero temperature and the integrable structure of the model, which is responsible for the discontinuous $D(\beta > 0)$ as a function of anisotropy, plays no role.

Next, we want to consider the high-temperature limit in more detail. In Fig. 2 the difference between $\beta^{-1}D(\beta)$ and the bound (10) is shown. We note first that for both sets of anisotropies, $\gamma = \pi/m$ and $\gamma = 7\pi/m$, the numerical data show a power-law decay in temperature towards the high-temperature bound. Interestingly, the Drude weights at a given temperature order differently as a function of anisotropy in the case $\gamma = \pi/m$ for temperatures above and below $\beta^{-1} \approx 0.5$.

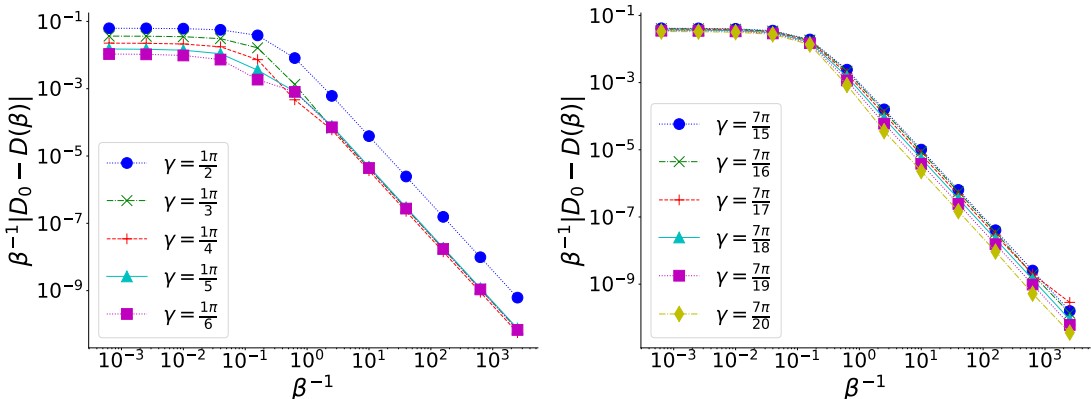

Figure 2: Absolute difference between the Prosen bound $\beta^{-1}D_{\beta\to 0}$ of (10) and the Drude weight coefficient $\beta^{-1}D$ demonstrating convergence to the Prosen bound. Note the crossover between curves for different anisotropies at $\beta^{-1} \approx 0.5$ in the left panel. A power-law scaling consistent with $|D(\beta)-D_{\beta\to 0}| \sim \beta^3$ at high temperatures is observed, agreeing with the TBA result see Eq. (52).

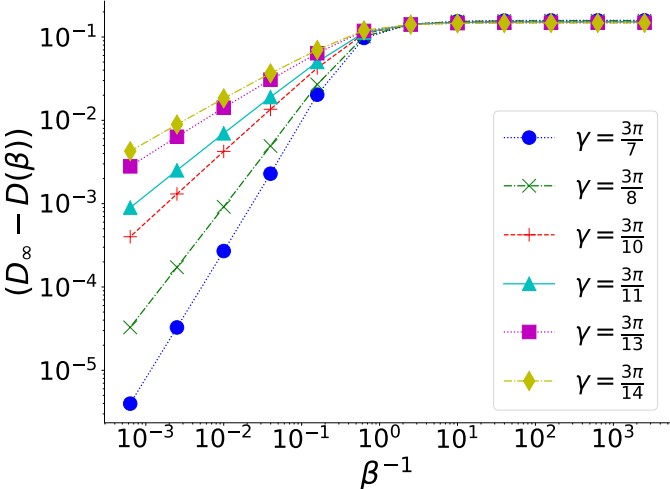

Figure 3: The numerical data at low temperatures show a power-law scaling consistent with $D_{\beta\to\infty} - D(\beta) \sim \beta^{2\gamma/(\gamma-\pi)}$.

Finally, we also want to consider the low-temperature limit for general anisotropies $\gamma = \pi n/m$. In Ref. [4] it was observed that the Drude weight at low temperatures scales as $D(\beta) \sim D_{\beta\to\infty} - \alpha\beta^{2/(1-m)}$ for anisotropies $\gamma = \pi/m$ with some constant $\alpha$. If this scaling does hold for all commensurate anisotropies then one would expect $D(\beta) \sim D_{\beta\to\infty} - \alpha\beta^{2\gamma/(\gamma-\pi)}$. In Fig. 3 we show exemplarily for anisotropies $\gamma = 3\pi/m$ that this expectation is consistent with our numerical data.

## 5 Conclusions

The main purpose of this paper was to provide a first principles derivation for the expectation value of the spin current as well as current-charge and charge-charge expectation values in

steady states described by given particle and hole distributions within the TBA approach. The main ingredient to derive exact formulas for these quantities was to relate the spin current with a matrix element of the two-site reduced density matrix. We then used the fact that this matrix element can be obtained from a generalized free energy by taking a derivative with respect to a spectral parameter while the charges were generated by taking derivatives with respect to the Lagrange parameters (generalized temperatures) $\beta_n$. We showed that the results derived in this way are consistent with a multi-particle generalization of known Lieb-Liniger results as conjectured in [15] and hence with the formula in [4]. Using the Mazur bound and assuming that it becomes exhausted if one considers the full TBA particle content we also derived a closed-form expression for the spin Drude weight. Straightforward manipulations showed that our result is identical to the TBA result obtained 20 years ago based on calculating the curvature of energy levels and using the finite-temperature Kohn formula. While consistent results for the Drude weight have now been obtained by the Kohn formula, by constructing the quasi-local charges protecting the Drude weight, and by the approach presented in this paper we would like to stress that all of these results are limited to the commensurate anisotropies $\gamma = \pi n/m$ and make use of the TBA formalism. While the construction of quasi-local charges has provided a definitive finite lower bound for anisotropies $|\Delta| < 1$ it is, in our view, still not completely excluded that parts of the Drude weight are missed in these approaches. In particular, it seems to us that we are not at the point yet where we can claim that a fractal structure of $D(\beta > 0)$ has been fully proven.

Putting such fundamental questions aside and taking the TBA result $D(\beta)$ as given, we have extended the analysis of the high-temperature asymptotics from the case $\gamma = \pi/m$ to all commensurate anisotropies $\gamma = \pi n/m$. Our analytical result in the high-temperature limit is identical to the bound obtained previously by considering the contribution of all known quasi-local charges. Finally, we have also presented a numerical evaluation of $D(\beta)$ for all temperatures showing that the TBA Drude weight has fractal character for all finite temperatures and that the low-temperature scaling follows a power law with exponent $D_{\beta \to \infty} - D(\beta) \sim \beta^{2\gamma/(\gamma-\pi)}$.

## Acknowledgements and funding information

The authors acknowledge support by the Deutsche Forschungsgemeinschaft (DFG) via Research Unit FOR 2316. JS acknowledges support by the Natural Sciences and Engineering Research Council (NSERC, Canada).

## A  Proof of identity (28)

The identity (28), which we want to prove here, can also be written as

$$\int d\theta\, \sigma q^k \partial_m \partial_n \ln(1 + \eta^{-1}) = \int d\theta\, \sigma \frac{\widetilde{q}^k \widetilde{q}^m \widetilde{q}^n}{(1 + \eta)(1 + \eta^{-1})}, \tag{53}$$

where the subscript $\ell$ is omitted, implicitly understanding the summation over it. We first use the fundamental BA equation (19) obtaining

$$\partial_k \ln \eta = q^k + K * \sigma \partial_k \ln(1 + \eta^{-1}), \quad \partial_m \partial_n \ln \eta = K * \sigma \partial_m \partial_n \ln(1 + \eta^{-1}), \tag{54}$$

and therefore

$$\int d\theta\, (\underbrace{\partial_k \ln \eta - q^k}_{K * \sigma \partial_k \ln(1+\eta^{-1})})\sigma \partial_m \partial_n \ln(1 + \eta^{-1}) = \int d\theta\, \partial_k \ln(1 + \eta^{-1})\partial_m \partial_n \ln \eta. \tag{55}$$

For the l.h.s. of Eq. (53) we thus find

$$(A.1)_{\text{l.h.s.}} = \int d\theta \, \sigma (\partial_k \ln \eta) \, \partial_m \partial_n \ln(1 + \eta^{-1}) - \int d\theta \, \sigma \partial_k \ln(1 + \eta^{-1}) \partial_m \partial_n \ln \eta. \tag{56}$$

Finally, we need to calculate the following derivatives

$$\partial_k \ln(1 + \eta^{-1}) = -\frac{\partial_k \ln \eta}{1 + \eta}, \quad \partial_m \partial_n \ln(1 + \eta^{-1}) = -\frac{\partial_m \partial_n \ln \eta}{1 + \eta} + \frac{(\partial_m \ln \eta)(\partial_n \ln \eta)}{(1 + \eta)(1 + \eta^{-1})}. \tag{57}$$

Plugging this into Eq. (56) then leads to

$$(A.1)_{\text{l.h.s.}} = \int d\theta \, \sigma \frac{(\partial_k \ln \eta)(\partial_m \ln \eta)(\partial_n \ln \eta)}{(1 + \eta)(1 + \eta^{-1})} = \int d\theta \, \sigma \frac{\widetilde{q}^k \widetilde{q}^m \widetilde{q}^n}{(1 + \eta)(1 + \eta^{-1})}, \tag{58}$$

which proves the relation (28).

# B   Elementary Identity $p_L = 1/n$

In order to prove this identity we need the definitions of the Takahashi-Suzuki (TS) integers from [27], which are collected below. TS integers are defined for $\gamma/\pi \in Q$ in terms of its continued fraction, which we notate as $\gamma/\pi = 1/p_0 \equiv [\nu_1, \ldots, \nu_\alpha]$ and say has length $\alpha$. As an example take $\gamma = 4\pi/9$ whose continued fraction will be $\gamma/\pi = 1/p_0 = 1/(2 + 1/4) \equiv [2, 4]$ with length 2. To make the notation consistent with the literature on the Bethe strings we identify our $n$ from $\gamma = n\pi/m$ with the TS integer $z_\alpha$ and our $m$ with $y_\alpha$, which coincide with the $\alpha$-th terms of Eq. (59)

$$z_\ell = z_{\ell-2} + \nu_\ell z_{\ell-1}, \qquad\qquad y_\ell = y_{\ell-2} + \nu_\ell y_{\ell-1}. \tag{59}$$

Rational TS numbers $p_\ell$ are obtained in terms of the above integers by

$$y_\ell = z_\ell p_0 + (-1)^\ell p_{\ell+1}, \quad \text{with } p_{\alpha+1} = 0. \tag{60}$$

By induction we can show that $p_\alpha = 1/z_\alpha$. The initial induction step for $\alpha = 1$ is trivial, as $\gamma = \pi/\nu_1$ has $p_1 = 1 = 1/1$, which follows from the definitions.

For our induction hypothesis we take $p_\alpha = 1/z_\alpha$ for a continued fraction $\{\gamma\} \equiv [\nu_1, \ldots, \nu_\alpha]$. Consider a second anisotropy with continued fraction $\gamma'/\pi = [\nu_1, \ldots, \nu_\alpha, \nu'_{\alpha+1}]$, whose integers are denoted $\{z'_\ell, y'_\ell, p'_\ell\}$ with $\ell \in \{0, 1, \ldots, \alpha+1\}$. By definition (59) we know that the $\gamma$ and $\gamma'$ TS integers $\{z_\ell, y_\ell\} = \{z'_\ell, y'_\ell\}_{\ell < \alpha+1}$ agree for the first $\alpha$ values.

Then beginning from Eq. (60) with index $i = \alpha$

$$\begin{aligned}
y_\alpha &= z_\alpha p'_0 + (-1)^\alpha p'_{\alpha+1}, \\
y'_{\alpha+1} - y_{\alpha-1} &= \nu'_{\alpha+1} p'_0 z_\alpha + (-1)^\alpha p'_{\alpha+1} \nu'_{\alpha+1}, \\
y'_{\alpha+1} - z_{\alpha-1} + (-1)^\alpha p_\alpha &= \nu'_{\alpha+1} p'_0 z_\alpha + (-1)^\alpha p'_{\alpha+1} \nu'_{\alpha+1}, \\
y'_{\alpha+1} - z_{\alpha-1} p_0 + (-1)^\alpha p_\alpha &= (z'_{\alpha+1} - z_{\alpha-1}) p'_0 + (-1)^\alpha \nu'_{\alpha+1} p'_{\alpha+1}.
\end{aligned} \tag{61}$$

With $p'_0 y_{\alpha+1} = z_{\alpha+1}$ the relation simplifies to

$$z_{\alpha-1} p'_0 - z_{\alpha-1} p_0 + (-1)^\alpha p_\alpha = (-1)^\alpha \nu'_{\alpha+1} p'_{\alpha+1}. \tag{62}$$

From the induction hypothesis $p_\alpha = 1/z_\alpha$ so obtain

$$z_\alpha z_{\alpha-1} p'_0 - z_\alpha z_{\alpha-1} p_0 + (-1)^\alpha = (-1)^\alpha z_\alpha \nu'_{\alpha+1} p'_{\alpha+1}. \tag{63}$$

With the relation $p'_0 z_\alpha = y_\alpha - (-1)^\alpha p'_{\alpha+1}$ and Eq. (59) it follows that

$$(-1)^\alpha = (-1)^\alpha z_{\alpha+1} p'_{\alpha+1}, \tag{64}$$

so conclude that $p'_{\alpha+1} = 1/z_{\alpha+1}$ and the identity is proven.

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
