# Peer review of "The spin Drude weight of the XXZ chain and generalized hydrodynamics"

_SciPost Physics, doi:SciPost Phys. 6, 005 (2019)_

## Round 1 · Referee Report · Anonymous · 2018-9-26

Strengths
- alternative, fully independent, derivation of an intriguing result in high-temperature many-body physics
- high level of rigour (very few assumptions needed)
Weaknesses
- slightly confusing presentation (can be improved)
Report
The manuscript presents an alternative derivation of the expectation value of the spin current, as well as the current-charge and the charge-charge expectation values needed for evaluation of the Mazur bound on the Drude weight, in the steady states given by densities of TBA quasiparticles. The technique provides a nice complementary approach to a recent calculation based on Generalised Hydrodynanics - GHD (Ref.[28]), and it is nice to see that the two approaches exactly match. Even more satisfactory (to my eyes), is the observation that the result exactly matches the "fractal" lower bound for the spin Drude weight at high temperatures (for commensurate anisotropies). I agree with the authors that this is not a proof of the fractal Drude weight, but it is certainly a strong additional evidence.
I don't want to get into a discussion which approach is "more rigorous", but the current one seems to need somewhat less assumptions than GHD, even though it is still not mathematically rigorous.
I think the paper is a valuable addition to an extensive literature on the subject and I strongly recommend its publication in SciPost.
Requested changes
I have the following remarks on the presentation which I would ask the authors to consider when preparing a final version:
1- Maybe ref [28] can be quoted already in Sect. 2 (after Eq. (2.10)) where a similarity of formulation to that of GHD is mentioned for the first time.
2- A remark on what exact charges Q_n are used in formula Eq. (2.16) and below, expressing the spin Drude weight in terms of Mazur bound. They can’t be the simple local ones introduced in Eq. (1.2), as, as has been argued, these have vanishing overlap with the spin current at vanishing magnetic field? Are Q_n the quasi-local charges, but then they have not been introduced in the text? Perhaps more explanation on this point is needed to avoid confusion (like mine).
3- Using symbols for both temperature (T) and inverse temperature (\beta) is somehow confusing. Perhaps one may only use \beta?
4- Is the finite-temperature correction to formula (3.21) really O(\beta^3)? I would find this surprising.. If so, the authors can perhaps comment on where it comes from?
Author: Jesko Sirker on 2018-11-21 [id 350]
(in reply to Report 1 on 2018-09-26)
1- Maybe ref [28] can be quoted already in Sect. 2 (after Eq. (2.10)) where a similarity of formulation to that of GHD is mentioned for the first time.
Reply: We checked the referee's suggestion and found that [28] would be misplaced. The reference is quoted already in Sect. 2 (after Eq. (2.10)).
2- A remark on what exact charges Q_n are used in formula Eq. (2.16) and below, expressing the spin Drude weight in terms of Mazur bound. They can't be the simple local ones introduced in Eq. (1.2), as, as has been argued, these have vanishing overlap with the spin current at vanishing magnetic field? Are Q_n the quasi-local charges, but then they have not been introduced in the text? Perhaps more explanation on this point is needed to avoid confusion (like mine).
Reply: This is indeed an interesting point and we thank the referee for this comment. From a practical perspective, one could take the point of view that it does not matter what the additional charges Q_1, Q_2, ... are as long as they make the set of charges complete. If this is the case, then the Mazur argument can be applied and the formula (3.6) follows in which all other charge densities have dropped out. The essential point to have a finite Drude weight is that the correlator <J_0 Q_n> is known and is nonzero. We therefore believe that a proper first principle derivation of this correlator as presented in our manuscript is an important step in making the result for the Drude weight more rigorous.
3- Using symbols for both temperature (T) and inverse temperature (\beta) is somehow confusing. Perhaps one may only use \beta?
Reply: See list of changes.
4- Is the finite-temperature correction to formula (3.21) really O(\beta^3)? I would find this surprising.. If so, the authors can perhaps comment on where it comes from?
Reply: This is a typo. See list of changes.
Author: Jesko Sirker on 2018-11-21 [id 351]
(in reply to Report 2 on 2018-10-03)However, the paper has many flaws, especially in some of its (too strong) claims and in the description of what has been done before. It is essential that these flaws be corrected before the paper can be published.
General reply: The approach presented in our manuscript is complementary and completely independent from the GHD approach. The Bethe-Boltzmann (or Euler equation if this is the term the referee prefers) central to the GHD approach are not used at all. Wherever appropriate, we compare with the GHD and cite accordingly.
The first main logical flaw is in the claim that the Drude weight derivation in section 3 is new and more complete / rigorous than the derivation presented in [15]. This is incorrect. In [15], the Drude weight was derived exactly from the projection formula (essentially eq 1.7, but more written without assumption of orthonormality) along with a completeness assumption, which is indeed the formula / assumption that the authors use (they use Mazurâs minimisation argument instead of the direct calculation of the projection, this is nice and interesting, but this evaluates the same projection formula under the same assumptions). The only conjecture used in the derivation [15], and not used in the present manuscript, was the exact current formula.
Reply: In section 3 we use indeed Mazur's (in)equality as outlined in Mazur's original paper and so does Ref. [15]. The Mazur paper is cited as is Ref. [15]. We do not claim that we have invented the Mazur (in)equality nor do the authors of Ref. [15]. We cannot see any 'main logical flaw' in what we present.
The authors here prove it for the spin current in section 2, which is a nice result. But other than that, no other conjecture is used in [15] that is not used here, in particular there is no need for coarse graining and ray-dependent local stationary states. The last sentence just before section 3.1, on page 8, is incorrect and should be deleted, and it should be stressed that the derivation presented in section 3 is equivalent to that presented in [15].
Reply: It is our understanding that the GHD approach is based on a coarse graining and that one cannot extract short-distance or short-time behavior from it. If the referee is aware of any results to the contrary we would be interested to learn about them. Once more, both the derivation of the Drude weight formula in [15] and in our manuscript use Mazur's result which is cited appropriately. The derivation in [15] is based on a conjectured form of <J_0 Q_n> while we derive this correlator from first principles.
Secondly, the equations 1.11 - 1.13 and the explanations given for generalised hydro are very sloppy. First, the notations Qn and Jn seem to be used for the total, space-integrated (site-summed) charges and currents (this should be made more clear in the paper). However, of course, the quantities that satisfy the continuity relation 1.11 are the densities, not the space-integrated quantities. This equation should be corrected.
Reply: See list of changes.
Second, the form 1.12, that the continuity relations are supposedly taking in GHD, is not correct. The proposal does not say that the local stationary states must depend on the ray xi = x/t. The local stationary states depend in general on both x and t. Please read the papers [12,13] carefully. Indeed many later works have used the full x,t-dependence of the local stationary states. The special dependence on x/t only comes when the initial state is scale invariant, which is the case for domain wall initial condition for instance.
Reply: See list of changes.
Third, I should also say that the terminology Bethe-Boltzmann - no widely used - is a bit misleading. The (corrected) equation 1.12 is a Euler equation, not a (collisionless) Boltzmann equation (which is just a Liouville equation). That is, it is not valid at small densities like a collisionless Boltzmann equation is; it is instead valid at large wavelengths like a Euler equation is.
Reply: See list of changes
Fourth, in eq. 1.13, The form of the quantities of charge carried by the quasiparticles seem to be very special: here the authors assume a particular choice of the charges Q_m with eigenvalues that are powers of q_l(theta). But this seems an inconsistent notation: the choice is not the one where orthogonality holds, and orthogonality is used in 1.7. This should be clarified.
Reply: These are not powers but rather upper indices. A short note has been added.
Finally, other small things should be corrected. In section 4, two numerical schemes are used to evaluate the Drude weight formula. The first is based on a linear response protocol. It should be pointed out that the equivalence of the Drude weight formula as presented eq 3.6 was already proven to be equivalent to the linear response scheme in [15; sect 5.1].
Reply: See list of changes.
Sentence 'This formula agrees with the conjectured general current formula used in GHD and appearing in Ref. [15]' page 6: in fact the general formula for current averages first appeared in ref [12,13]. It was conjectured in XXZ in ref [12] with very strong numerical checks, and it was proven using crossing symmetry in relativistic QFT in ref [13].
Sentence 'The GHD conjectures are thus proved both for J0 and for J1 = JE' page 7: For J1 = JE the conjecture was already proven - as it is indeed a trivial application of TBA (it is immediate from the formulae of [12,13] for instance). I don't think the author should claim they are the first to provide the proof.
Reply: It seems that the formula for JE was first given explicitly in Ref. [2] which is cited. If one wants to say that it immediately follows in TBA anyhow, then one should correctly say that it follows from expressions given in [4] which was published much earlier than [12,13]. Furthermore, we insist that it is important to clearly distinguish between a conjecture (if the numerical data would be in contradiction then it would obviously no longer be a valid conjecture) and a proof from first principles. We are discussing lattice models here, not relativistic QFT's.
Formula 3.1, first equality is only valid for Qn orthogonal basis, and second equality is incorrect because the particular choice of Qn (with powers of ql) is not orthogonal. (The derivation given afterwards is correct however, as it uses Mazur's argument.)
Reply: There are no powers here. The superscript n and the subscript l are just indices. Some comments have been added, see list of changes.
Conclusion, page 12: note that the Lieb-Linger proposal given in [15] was already conjectured there to be applicable in the multiparticle case, and it was mentioned to agree with the 20-years-old results (see the introduction of [15]).
Reply: See list of changes.
Please define more precisely what Ekin is in eq 1.5.
Reply: See list of changes.
List of changes:
-Changed all temperatures '$T$' to inverse temperatures '$\beta$' (also the references to temperature in certain places)
-Eq. 1.5 E_kin to H_kin.
-1.11-1.12 changed to lower-case q and j and added sentence above 1.12.
-1.12: Footnote regarding Euler versus Bethe-Boltzmann Eq. added.
-Sentence around 1.13 to clarify the notation (ie. draw attention to the superscript).
-Below Eq. 2.18 clarified that this is not the first derivation of the energy current.
-Below Eq. 3.4 added a paragraph clarifying what is meant by completeness and what the role of the additional conserved charges is.
-added footnote on page 7: Taking <J0 J0> as shorthand for lim <J0(0)J0(t)>
-Below Eq. 3.8 we now point out that the derivation relies on completeness and TBA, so is not to be understood as completely independent
-Corrected typo Eq. 3.21 (ie. O(\beta^2) is the next surviving term as 3.21 is written)
-In section 4 (1st paragraph) point out the equivalence of linear response and GHD. Citation to Ref. [15] added.
-Added description under figure 2 to clarify what power law was observed.
-Mention conjectured multi-particle formula in Ref. [15] in the conclusions

---

## Round 1 · Referee Report · Anonymous · 2018-10-3

Strengths
new and rigorous derivation of exact spin current in the XXZ model.
interesting derivation of the Drude weight.
Weaknesses
Some part of the literature is not overviewed quite correctly.
Some claims about what is new are too strong
Report
In this paper, the authors provide a first-principle proof of the general expression of the spin current in GGEs of the XXZ chain. They then use this result to derive expressions for density-current and current-current space-integrated correlators, and for the spin Drude weight. They then analyse the resulting Drude weight in various limits, providing a full large-temperature expression valid at general rational anisotropies and some numerical evaluations of the formula.\[\]
I think the paper is interesting, in that, I believe, this is the first first-principle calculation of the spin current in GGEs of the XXZ chain. The main observations are equations 2.1 and 2.2; equation 2.1 was already obtained quite some time ago, but it is a good (if rather simple) observation that this leads to the exact spin current - to my knowledge, this observation had not been made before (but this may be my lack of knowledge). The authors then go on to show that this expression agrees with proposals made recently in the context of generalised hydrodynamics. The conjecture in XXZ had never been proven, for any nontrivial current (this excludes, of course, the energy current, which is itself a conserved density), and it is an important problem to establish this. Hence the result of the paper is interesting from this viewpoint.\[\]
However, the paper has many flaws, especially in some of its (too strong) claims and in the description of what has been done before. It is essential that these flaws be corrected before the paper can be published.\[\]
The first main logical flaw is in the claim that the Drude weight derivation in section 3 is new and more complete / rigorous than the derivation presented in [15]. This is incorrect. In [15], the Drude weight was derived exactly from the projection formula (essentially eq 1.7, but more written without assumption of orthonormality) along with a completeness assumption, which is indeed the formula / assumption that the authors use (they use Mazur’s minimisation argument instead of the direct calculation of the projection, this is nice and interesting, but this evaluates the same projection formula under the same assumptions). The only conjecture used in the derivation [15], and not used in the present manuscript, was the exact current formula. The authors here prove it for the spin current in section 2, which is a nice result. But other than that, no other conjecture is used in [15] that is not used here, in particular there is no need for coarse graining and ray-dependent local stationary states. The last sentence just before section 3.1, on page 8, is incorrect and should be deleted, and it should be stressed that the derivation presented in section 3 is equivalent to that presented in [15].\[\]
Secondly, the equations 1.11 - 1.13 and the explanations given for generalised hydro are very sloppy.\[\]
First, the notations $Q_n$ and $J_n$ seem to be used for the total, space-integrated (site-summed) charges and currents (this should be made more clear in the paper). However, of course, the quantities that satisfy the continuity relation 1.11 are the densities, not the space-integrated quantities. This equation should be corrected.\[\]
Second, the form 1.12, that the continuity relations are supposedly taking in GHD, is not correct. The proposal does not say that the local stationary states must depend on the ray $\xi=x/t$. The local stationary states depend in general on both $x$ and $t$. Please read the papers [12,13] carefully. Indeed many later works have used the full $x,t$-dependence of the local stationary states. The special dependence on $x/t$ only comes when the initial state is scale invariant, which is the case for domain wall initial condition for instance.\[\]
Third, I should also say that the terminology “Bethe-Boltzmann” - no widely used - is a bit misleading. The (corrected) equation 1.12 is a Euler equation, not a (collisionless) Boltzmann equation (which is just a Liouville equation). That is, it is not valid at small densities like a collisionless Boltzmann equation is; it is instead valid at large wavelengths like a Euler equation is.\[\]
Fourth, in eq. 1.13, The form of the quantities of charge carried by the quasiparticles seem to be very special: here the authors assume a particular choice of the charges Q_m with eigenvalues that are powers of q_l(theta). But this seems an inconsistent notation: the choice is not the one where orthogonality holds, and orthogonality is used in 1.7. This should be clarified.\[\]
Finally, other small things should be corrected.\[\]
In section 4, two numerical schemes are used to evaluate the Drude weight formula. The first is based on a linear response protocol. It should be pointed out that the equivalence of the Drude weight formula as presented eq 3.6 was already proven to be equivalent to the linear response scheme in [15; sect 5.1].\[\]
Sentence “This formula agrees with the conjectured general current formula used in GHD and appearing in Ref. [15]” page 6: in fact the general formula for current averages first appeared in ref [12,13]. It was conjectured in XXZ in ref [12] with very strong numerical checks, and it was proven using crossing symmetry in relativistic QFT in ref [13].\[\]
Sentence “The GHD conjectures are thus proved both for J0 and for J1 = JE” page 7: For $J_1$ = $J_E$ the conjecture was already proven - as it is indeed a trivial application of TBA (it is immediate from the formulae of [12,13] for instance). I don’t think the author should claim they are the first to provide the proof.\[\]
Formula 3.1, first equality is only valid for $Q_n$ orthogonal basis, and second equality is incorrect because the particular choice of $Q_n$ (with powers of $q_\ell$) is not orthogonal. (The derivation given afterwards is correct however, as it uses Mazur’s argument.)\[\]
Conclusion, page 12: note that the Lieb-Linger proposal given in [15] was already conjectured there to be applicable in the multiparticle case, and it was mentioned to agree with the 20-years-old results (see the introduction of [15]).\[\]
Please define more precisely what $E_{\rm kin}$ is in eq 1.5.
Requested changes
1. The last sentence just before section 3.1, on page 8, is incorrect and should be deleted, and it should be stressed that the derivation presented in section 3 is equivalent to that presented in [15], except for the only point that here, the spin current is properly derived, while in [15] it was taken from the conjectured expression stemming from the GHD works [12,13].
2. Eqs. 1.11 and 1.12 should be corrected.
3. Emphasize that 1.12 is a Euler equation (not a Boltzmann equation)
4. Make more consistent form 1.13 of conserved quantities (which are not orthogonal)| with relation 1.7 that involves orthogonality
5. mention that equivalence of numerical schemes of section is shown in [15]
6. correctly cite [12,13] (page 6) for where current formula first appeared
7. mention that formula for j_E (page 7) is not a first-time proof
8. correct equation 3.1
9. adjust conclusion as per above comment
10. define E_kin in eq 1.5

---

## Round 2 · Referee Report · Anonymous (Referee 2) · 2018-12-11

Report

I thank the authors for having clarified most aspects. The calculations now seem to be clear and all correct, and I reiterte that the result is interesting.

I agree that the result for the energy current average appeared much before [12,13]. Ref 2 has it explicitly, while Ref 4 might be a more correct earlier reference. Probably the correct earliest reference is Takahashi's paper on the TBA of XXZ, as the formula - at least in thermal states - follows immediately from the free energy. Maybe comment on these papers on page 7 just before sect 3.

I still think there is confusion concerning the results obtained in the context of GHD, in the description of the various methods on pages 3-4. I was probably not as clear as I should have been in my previous report. There are three main references where Drude weights are evaluated following GHD developments: refs 14, 28 and 15. Earliest are refs 14 and 28, doing XXZ (but the derivation is applicable to other models). There the exact solution for nonequilibrium currents in the domain-wall problem (obtained in refs 12,13) is used, along with linear response. This exact solution is based on Euler GHD equations, and so uses the coarse-graining hypothesis, and the derivation of the Drude weight from it is based on a linear response assumption. These indeed fit within what the authors refer to as method 3, and should be mentioned there. Slightly later, ref 15 provided a general formula in the Lieb-Liniger model, derived in a different way (again the derivation is applicable to other models). It does not use coarse graining, or linear response, or equations 1.11 or 1.12 of the present manuscript. It is not based on Euler equations, and no reference is made to the domain-wall problem. It is based on method 2, summing over conserved charges, overlapping with the current, exactly like the method of the present manuscript. The only part of the GHD developments that Ref 15 used, in deriving the Drude weight, is the conjectured expression for currents in arbitrary stationary states (GGEs); these expressions were not known before. Once this formula is known, the required correlation functions on the rhs of 1.7, not just at finite temperatures but in any GGE, follow just by differentiation, and the Drude weight can be evaluated, as done in ref 15. So I think ref 15 should be put in method 2, not method 3.

From this perspective, the main development of the present manuscript - an important development - is the proof of the exact spin current expression in arbitrary stationary states (GGEs). These are then used, much like in ref 15 (but using Mazur's method instead), to derive the Drude weight.

So, the sentence, page 4, starting as "For the Lieb-Liniger model in the linear response regime, in particular, this formalism has been used to obtain formulas for the expectation values of ..." is not accurate. Similarly, in saying the approach presented in the manuscript is complementary to that of GHD, one should bear in mind the above.

I would like the authors to make small adjustments on pages 3-4 to account for the above.

Finally, I still don't see why the authors talk about rays in: "If one assumes that a system which is not in equilibrium is composed of cells which are locally described by the distribution...". The cell-decomposition assumption is already in eq 1.12, and leads to ray-dependent states only in the domain-wall (or similar) problems. I would just take away the discussion of rays - unless the authors want to give more explanations about the domain-wall initial condition problem and the Drude weight calculation presented in refs 14, 28.

Requested changes

1- small changes in pages 3, 4 to account for contributions of refs 14,15,28 as described above.

2- take away discussion of rays on page 4, or described more at length the domain-wall initial value problem for GHD in the context of describing the results of refs 14, 28.

3- maybe add comments on ealier references for energy current in XXZ on page 7 just before sect 3.

---

## Round 2 · Author Response

First report:

However, the paper has many flaws, especially in some of its (too strong) claims and in the description of what has been done before. It is essential that these flaws be corrected before the paper can be published.

General reply: The approach presented in our manuscript is complementary and completely independent from the GHD approach. The Bethe-Boltzmann (or Euler equation if this is the term the referee prefers) central to the GHD approach are not used at all. Wherever appropriate, we compare with the GHD and cite accordingly.

The first main logical flaw is in the claim that the Drude weight derivation in section 3 is new and more complete / rigorous than the derivation presented in [15]. This is incorrect. In [15], the Drude weight was derived exactly from the projection formula (essentially eq 1.7, but more written without assumption of orthonormality) along with a completeness assumption, which is indeed the formula / assumption that the authors use (they use Mazur’s minimisation argument instead of the direct calculation of the projection, this is nice and interesting, but this evaluates the same projection formula under the same assumptions). The only conjecture used in the derivation [15], and not used in the present manuscript, was the exact current formula.

Reply: In section 3 we use indeed Mazur's (in)equality as outlined in Mazur's original paper and so does Ref. [15]. The Mazur paper is cited as is Ref. [15]. We do not claim that we have invented the Mazur (in)equality nor do the authors of Ref. [15]. We cannot see any 'main logical flaw' in what we present.

The authors here prove it for the spin current in section 2, which is a nice result. But other than that, no other conjecture is used in [15] that is not used here, in particular there is no need for coarse graining and ray-dependent local stationary states. The last sentence just before section 3.1, on page 8, is incorrect and should be deleted, and it should be stressed that the derivation presented in section 3 is equivalent to that presented in [15].

Reply: It is our understanding that the GHD approach is based on a coarse graining and that one cannot extract short-distance or short-time behavior from it. If the referee is aware of any results to the contrary we would be interested to learn about them. Once more, both the derivation of the Drude weight formula in [15] and in our manuscript use Mazur's result which is cited appropriately. The derivation in [15] is based on a conjectured form of <J_0 Q_n> while we derive this correlator from first principles.

Secondly, the equations 1.11 - 1.13 and the explanations given for generalised hydro are very sloppy. First, the notations Qn and Jn seem to be used for the total, space-integrated (site-summed) charges and currents (this should be made more clear in the paper). However, of course, the quantities that satisfy the continuity relation 1.11 are the densities, not the space-integrated quantities. This equation should be corrected.

Reply: See list of changes.

Second, the form 1.12, that the continuity relations are supposedly taking in GHD, is not correct. The proposal does not say that the local stationary states must depend on the ray xi = x/t. The local stationary states depend in general on both x and t. Please read the papers [12,13] carefully. Indeed many later works have used the full x,t-dependence of the local stationary states. The special dependence on x/t only comes when the initial state is scale invariant, which is the case for domain wall initial condition for instance.

Reply: See list of changes.

Third, I should also say that the terminology Bethe-Boltzmann - no widely used - is a bit misleading. The (corrected) equation 1.12 is a Euler equation, not a (collisionless) Boltzmann equation (which is just a Liouville equation). That is, it is not valid at small densities like a collisionless Boltzmann equation is; it is instead valid at large wavelengths like a Euler equation is.

Reply: See list of changes

Fourth, in eq. 1.13, The form of the quantities of charge carried by the quasiparticles seem to be very special: here the authors assume a particular choice of the charges Q_m with eigenvalues that are powers of q_l(theta). But this seems an inconsistent notation: the choice is not the one where orthogonality holds, and orthogonality is used in 1.7. This should be clarified.

Reply: These are not powers but rather upper indices. A short note has been added.

Finally, other small things should be corrected. In section 4, two numerical schemes are used to evaluate the Drude weight formula. The first is based on a linear response protocol. It should be pointed out that the equivalence of the Drude weight formula as presented eq 3.6 was already proven to be equivalent to the linear response scheme in [15; sect 5.1].

Reply: See list of changes.

Sentence 'This formula agrees with the conjectured general current formula used in GHD and appearing in Ref. [15]' page 6: in fact the general formula for current averages first appeared in ref [12,13]. It was conjectured in XXZ in ref [12] with very strong numerical checks, and it was proven using crossing symmetry in relativistic QFT in ref [13].

Sentence 'The GHD conjectures are thus proved both for J0 and for J1 = JE' page 7: For J1 = JE the conjecture was already proven - as it is indeed a trivial application of TBA (it is immediate from the formulae of [12,13] for instance). I don't think the author should claim they are the first to provide the proof.

Reply: It seems that the formula for JE was first given explicitly in Ref. [2] which is cited. If one wants to say that it immediately follows in TBA anyhow, then one should correctly say that it follows from expressions given in [4] which was published much earlier than [12,13]. Furthermore, we insist that it is important to clearly distinguish between a conjecture (if the numerical data would be in contradiction then it would obviously no longer be a valid conjecture) and a proof from first principles. We are discussing lattice models here, not relativistic QFT's.

Formula 3.1, first equality is only valid for Qn orthogonal basis, and second equality is incorrect because the particular choice of Qn (with powers of ql) is not orthogonal. (The derivation given afterwards is correct however, as it uses Mazur's argument.)

Reply: There are no powers here. The superscript n and the subscript l are just indices. Some comments have been added, see list of changes.

Conclusion, page 12: note that the Lieb-Linger proposal given in [15] was already conjectured there to be applicable in the multiparticle case, and it was mentioned to agree with the 20-years-old results (see the introduction of [15]).

Reply: See list of changes.

Please define more precisely what Ekin is in eq 1.5.

Reply: See list of changes.

Second report:

1- Maybe ref [28] can be quoted already in Sect. 2 (after Eq. (2.10)) where a similarity of formulation to that of GHD is mentioned for the first time.

Reply: We checked the referee's suggestion and found that [28] would be misplaced. The reference is quoted already in Sect. 2 (after Eq. (2.10)).

2- A remark on what exact charges Q_n are used in formula Eq. (2.16) and below, expressing the spin Drude weight in terms of Mazur bound. They can't be the simple local ones introduced in Eq. (1.2), as, as has been argued, these have vanishing overlap with the spin current at vanishing magnetic field? Are Q_n the quasi-local charges, but then they have not been introduced in the text? Perhaps more explanation on this point is needed to avoid confusion (like mine).

Reply: This is indeed an interesting point and we thank the referee for this comment. From a practical perspective, one could take the point of view that it does not matter what the additional charges Q_1, Q_2, ... are as long as they make the set of charges complete. If this is the case, then the Mazur argument can be applied and the formula (3.6) follows in which all other charge densities have dropped out. The essential point to have a finite Drude weight is that the correlator <J_0 Q_n> is known and is nonzero. We therefore believe that a proper first principle derivation of this correlator as presented in our manuscript is an important step in making the result for the Drude weight more rigorous.

3- Using symbols for both temperature (T) and inverse temperature (\beta) is somehow confusing. Perhaps one may only use \beta?

Reply: See list of changes.

4- Is the finite-temperature correction to formula (3.21) really O(\beta^3)? I would find this surprising.. If so, the authors can perhaps comment on where it comes from?

Reply: This is a typo. See list of changes.

---

## Round 2 · List of Changes

List of changes:

-Changed all temperatures `T' to inverse temperatures `\beta' (also the
references to temperature in certain places)

-Eq. 1.5 E_kin to H_kin.

-1.11-1.12 changed to lower-case q and j and added sentence above 1.12.

-1.12: Footnote regarding Euler versus Bethe-Boltzmann Eq. added.

-Sentence around 1.13 to clarify the notation (ie. draw attention to the
superscript).

-Below Eq. 2.18 clarified that this is not the first derivation of the energy
current.

-Below Eq. 3.4 added a paragraph clarifying what is meant by
completeness and what the role of the additional conserved charges is.

-added footnote on page 7: Taking <J0 J0> as shorthand for lim <J0(0)J0(t)>

-Below Eq. 3.8 we now point out that the derivation relies on completeness and
TBA, so is not to be understood as completely independent

-Corrected typo Eq. 3.21 (ie. O(\beta^2) is the next surviving term as 3.21 is
written)

-In section 4 (1st paragraph) point out the equivalence of linear response and
GHD. Citation to Ref. [15] added.

-Added description under figure 2 to clarify what power law was observed.

-Mention conjectured multi-particle formula in Ref. [15] in the conclusions

---

## Editorial Decision

published